# Saliva Testing Is Accurate for Early-Stage and Presymptomatic COVID-19

Abigail J. Johnson,[a] Shannon Zhou,[b] Susan L. Hoops,[c] Benjamin Hillmann,[c] Matthew Schomaker,[d] Robyn Kincaid,[d] Jerry Daniel,[e] Kenneth Beckman,[f] Daryl M. Gohl,[e,g] Sophia Yohe,[f] Dan Knights,[b,c] Andrew C. Nelson[f]

[a]Division of Epidemiology and Community Health, School of Public Health, University of Minnesota, Minneapolis, Minnesota, USA
[b]BioTechnology Institute, College of Biological Sciences, University of Minnesota, Saint Paul, Minnesota, USA
[c]Department of Computer Science and Engineering, University of Minnesota, Minneapolis, Minnesota, USA
[d]M Health Fairview, Minneapolis, Minnesota, USA
[e]University of Minnesota Genomics Center, University of Minnesota, Minneapolis, Minnesota, USA
[f]Department of Laboratory Medicine and Pathology, Division of Molecular Pathology and Genomics, University of Minnesota, Minneapolis, Minnesota, USA
[g]Department of Genetics, Cell Biology, and Development, University of Minnesota, Minneapolis, Minnesota, USA

Andrew C. Nelson and Dan Knights contributed equally to this article. Senior author order was determined by mutual agreement in context of significant contributions to conceptualization, funding acquisition, resource acquisition, data curation, supervision, and writing by both authors.

**ABSTRACT** Although nasopharyngeal samples have been considered the gold standard for COVID-19 testing, variability in viral load across different anatomical sites could cause nasopharyngeal samples to be less sensitive than saliva or nasal samples in certain cases. Self-collected samples have logistical advantages over nasopharyngeal samples, making them amenable to population-scale screening. To evaluate sampling alternatives for population screening, we collected nasopharyngeal, saliva, and nasal samples from two cohorts with varied levels and types of symptoms. In a mixed cohort of 60 symptomatic and asymptomatic participants, we found that saliva had 88% concordance with nasopharyngeal samples when tested in the same testing lab ($n = 41$) and 68% concordance when tested in different testing labs ($n = 19$). In a second cohort of 20 participants hospitalized for COVID-19, saliva had 74% concordance with nasopharyngeal samples tested in the same testing lab but detected virus in two participants that tested negative with nasopharyngeal samples on the same day. Medical record review showed that the saliva-based testing sensitivity was related to the timing of symptom onset and disease stage. We find that no sample site will be perfectly sensitive for COVID-19 testing in all situations, and the significance of negative results will always need to be determined in the context of clinical signs and symptoms. Saliva retained high clinical sensitivity for early-stage and presymptomatic COVID-19 while allowing easier collection, minimizing the exposure of health care workers, and need for personal protective equipment and making it a viable option for population-scale testing.

**IMPORTANCE** Methods for COVID-19 detection are necessary for public health efforts to monitor the spread of disease. Nasopharyngeal samples have been considered the best approach for COVID-19 testing. However, alternative samples like self-collected saliva offer advantages for population-scale screening. Meta-analyses of recent studies suggest that saliva is useful for detecting SARS-CoV-2; however, differences in disease prevalence, sample collection, and analysis methods still confound strong conclusions on the utility of saliva compared to nasopharyngeal samples. Here, we find that the sensitivity of saliva testing is related to both the timing of the sample collection relative to symptom onset and the disease stage. Importantly, several clinical vignettes in our cohorts highlight the challenges of medical decision making with limited knowledge of the associations between laboratory test data and the natural biology of infection.

Address correspondence to Andrew C. Nelson, nels2055@umn.edu.

**KEYWORDS** COVID-19, SARS-CoV-2, saliva, testing, symptoms

Throughout the COVID-19 pandemic, the spread of infection has significantly outpaced laboratory testing to identify severe acute respiratory syndrome coronavirus 2 (SARS-CoV-2). Seroprevalence studies performed in March to May of 2020 in the United States suggested that the number of infections was 10-fold greater than confirmed laboratory diagnoses (1). This scenario significantly challenged public health efforts to monitor and contain the spread of disease. Nasopharyngeal samples have been considered the gold standard for COVID-19 testing. However, alternative samples, like self-collected saliva, offer advantages for population-scale screening and may perform well in specific clinical situations.

A number of studies comparing saliva, oral, and/or nasal samples with nasopharyngeal samples have reported heterogeneity in sensitivity or positive percent agreement (PPA), ranging from 66% to 98% (2–12); this heterogeneous performance is likely impacted by differences in patient populations and methods of sample collection and processing. For example, optimization of saliva sample processing within one institution improved the performance of this sample type across two sequential studies (3, 5). More importantly, clinical test performance is dependent on preanalytic variables, such as collection timing relative to the patient's disease course and anatomic site of collection. A study of inpatients at an advanced disease stage showed that lower respiratory samples (bronchoalveolar lavage) were more frequently positive (93%) than pharyngeal (32%) or nasal (63%) samples (13). Further studies of samples from different anatomic sites at different points in disease course are necessary to better understand how these variables impact clinical test performance.

Here, we acquired patient-collected saliva and anterior nasal research specimens for comparison with concurrent provider-collected nasopharyngeal samples in both outpatient and inpatient settings. The clinical context of specimen collection, the timing of sample collection during disease course, and the analytical performance of different molecular tests were assessed for their impact on the test result agreement between different anatomic sites.

## RESULTS

**Study design and cohort definition.** Two distinct patient cohorts were included in the study (Fig. 1). Cohort 1 consisted of 354 patients with clinical nasopharyngeal results from tests collected in outpatient or emergency department settings. Cohort 1 study participants, presenting with both symptomatic and asymptomatic concerns for COVID-19, were enrolled opportunistically from a population receiving a nasopharyngeal COVID-19 test in outpatient screening or emergency department (ED) settings. Due to limited testing resources available during the study, we relied on nasopharyngeal results from routine clinical testing. Nasal and saliva samples were prospectively collected and biobanked for retrospective testing (see Materials and Methods). Analysis per design was performed on 30 positive and 30 negative samples, in accordance with the recommended clinical evaluation guidelines in the United States Food and Drug Administration (FDA) Policy for Coronavirus Disease-2019 Tests During the Public Health Emergency. After 30 positive cases were detected by clinical nasopharyngeal samples, enrollment was stopped and 30 negative cases were randomly selected for comparison.

Due to clinical test triage patterns outside the study's purview, clinical nasopharyngeal samples were routed to different test platforms within the hospital system. We divided cohort 1 into two groups to reflect testing methods used for the clinical nasopharyngeal test. Cohort 1A ($n = 41$, outpatient) had clinical nasopharyngeal samples and biobanked saliva and nasal samples tested on our Clinical Laboratory Improvement Amendments-certified laboratory-developed test (CLIA-LDT). Cohort 1B ($n = 19$, ED) had clinical nasopharyngeal sample testing performed on one of two different commercial testing platforms (see details in Materials and Methods and the supplemental material). All saliva and nasal samples from cohort 1 were prospectively collected, biobanked, and then tested on the CLIA-LDT.

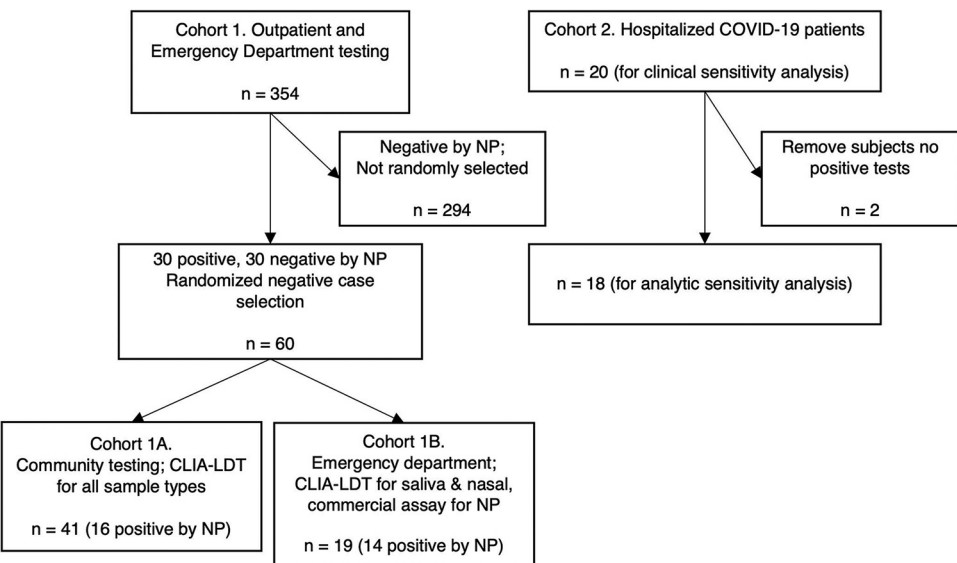

**FIG 1** Cohort flow diagram. Cohort 1 was enrolled prospectively at outpatient screening and emergency department locations for sample collection and biobanking. When the prespecified 30 positive samples were acquired, retrospective testing of saliva and anterior nasal samples was performed for 30 positives and 30 randomly selected negative samples. Heterogeneity in clinical nasopharyngeal (NP) test routing during the pandemic response necessitated subcohort analysis of cohorts 1A and 1B. Cohort 2 was enrolled and tested prospectively at an inpatient ward dedicated to COVID-19 patient care. Clinical sensitivity analysis is performed on all 20 patients enrolled in Cohort 2, while 2 patients were excluded from analytic sensitivity analysis due to negative test results for all three anatomic sites.

Cohort 2 consisted of 20 inpatients admitted to a dedicated COVID-19 ward for symptomatic disease management following prior positive laboratory testing for SARS-CoV-2 (range, 1 day to 4 weeks prior to admission). In this group, nasopharyngeal, saliva, and nasal samples were obtained at the same time from participants for concurrent analysis on the CLIA-LDT. Both cohorts were recruited, consented, and enrolled under protocols approved by the University of Minnesota's Institutional Review Board (cohorts 1A and 1B, STUDY00009393; cohort 2, STUDY00009560).

**Cohort characteristics.** Cohort 1A had an average age of 39.6 years and was skewed toward female gender, at 92.7%. Cohort 1B had an average age of 44.5 years and was skewed toward male gender, at 63.2%. Reliable clinical notes to establish symptoms were available for 18 of 41 patients in cohort 1A and 16 of 19 patients in 1B. Symptomatic patients comprised approximately 80% of evaluable patients in both cohorts 1A ($n = 14/18$) and 1B ($n = 13/16$), although a higher proportion of symptomatic patients in cohort 1B had positive clinical test results (92.3%) versus 1A (64.3%). Four asymptomatic patients in cohort 1A had negative clinical tests. Of 3 asymptomatic patients in cohort 1B, one had a positive clinical test and two had negative clinical tests. Cohort 2 had an average age of 62.5 years, consistent with the increased hospitalization rate of older COVID-19 patients. Gender was equally distributed in cohort 2 (Table 1).

**Concordance of testing on saliva, nasal, and nasopharyngeal samples.** Clinical nasopharyngeal testing results in cohort 1A ($n = 41$) identified 16 positive and 25 negative patients. Testing of banked saliva and nasal samples showed 87.5% PPA and 95.1% overall percent agreement (OPA), respectively, compared to nasopharyngeal samples (Table 2; see also Table S1 in the supplemental material). Saliva and nasal results were 100% concordant in cohort 1A.

Clinical nasopharyngeal testing on two different commercial platforms in cohort 1B ($n = 19$) produced 14 positive and 5 negative results. Testing banked saliva and nasal samples on the CLIA-LDT provided more discordant results: the PPA for both saliva and nasal samples against NP was 57.1%. This comparison is subject to platform bias. However, separate internal quality assurance data (see the supplemental material)

**TABLE 1** Cohort characteristics[a]

| Characteristic | Value for cohort: | | |
|---|---|---|---|
| | 1A | 1B | 2 |
| $n$ | 41 | 19 | 20 |
| Age (yr, $\pm$SD) | 39.6 $\pm$ 10.9 | 44.5 $\pm$ 20.1 | 62.5 $\pm$ 18.1 |
| Sex (% female) | 92.7 | 36.8 | 50.0 |
| % Participants with symptoms | 77.8 | 81.3 | 100.0 |
| % Participants with symptoms who are positive | 64.3 | 92.3 | 90.0 |
| % Participants without symptoms who are positive | 0.0 | 33.3 | 0.0 |
| N1_NP tests with $C_T$ values ($n$) | 16 | 14 | 16 |
| N1_Nasal tests with $C_T$ values ($n$) | 14 | 8 | 11 |
| N1_Saliva tests with $C_T$ values ($n$) | 13 | 8 | 12 |
| N2_NP tests with $C_T$ values ($n$) | 16 | 14 | 16 |
| N2_Nasal tests with $C_T$ values ($n$) | 13 | 7 | 11 |
| N2_Saliva tests with $C_T$ values ($n$) | 13 | 8 | 14 |

[a]N1 and N2 refer to primer-probe sets for the SARS-CoV-2 N gene (N1 and N2 targets). NP, nasal pharyngeal; $C_T$, cycle threshold.

showed 90% and 97% positive agreement of archived nasopharyngeal samples comparing each of the two commercial assays to the CLIA-LDT (Tables S4 and S5), suggesting that platform bias is not the only reason for the level of discordance observed across anatomic sample types in cohort 1B. Further, there were two discrepancies between nasal and saliva sample results (both tested on the CLIA-LDT) producing 87.5% PPA and 89.5% OPA between these sample types (Table S2).

Simultaneous collection and immediate testing of all three sample types for cohort 2 on the CLIA-LDT demonstrated 16 positive nasopharyngeal, 14 positive saliva, and 11 positive nasal samples from 20 patients. The PPA ranged from 69% to 82% and the OPA from 65% to 75% for all pairwise comparisons (Table 2). Saliva sampling identified 2 positive patients who tested negative by nasopharyngeal sampling, and inversely nasopharyngeal testing identified 4 positive patients who tested negative by saliva (Table S3). Nasal samples performed poorly in this patient cohort.

We used a composite reference standard approach to assess diagnostic sensitivity in cohort 2 (14, 15). All patients positive for SARS-CoV-2 by at least one upper respiratory sample type ($n = 18$) were considered a reference positive case for the assessment of sensitivity. Using this approach, the diagnostic sensitivity for each sample type was 89% for nasopharyngeal, 78% for saliva, and 61% for nasal.

**Comparison of cycle threshold results across sample type.** Cycle threshold ($C_T$) values for each anatomic sample site were compared in cohorts 1A and 2, and all samples were analyzed on the same analytical platform. The interquartile ranges for both N1 and N2 were largely overlapping (Fig. 2A, Fig. S1A). In cohort 1A, the 2 discordant nasopharyngeal sample data points had $C_T$ values of 36.6 and 38.4, near the assay limit of detection. Cohort 2 had more discordant data points, consistent with the higher average $C_T$ value (lower relative viral load) observed in all anatomic sample types in this inpatient cohort tested at later disease stages. Discordant data points in this cohort ranged as low as 26.2, suggesting greater biologic variability in relative viral load between anatomic sites in patients at later disease stages. Similar results were observed with relative viral load (Fig. S2).

**TABLE 2** PPA and OPA for samples[a]

| Cohort | % PPA and OPA for: | | | | | |
|---|---|---|---|---|---|---|
| | Nasal versus NP | | Saliva versus NP | | Saliva versus nasal | |
| | PPA | OPA | PPA | OPA | PPA | OPA |
| 1A ($n = 41$) | 87.5 | 95.1 | 87.5 | 95.1 | 100.0 | 100.0 |
| 2 ($n = 20$) | 68.8 | 75.0 | 75.0 | 70.0 | 81.8 | 65.0 |
| 1B ($n = 19$) | 57.1 | 68.4 | 57.1 | 68.4 | 87.5 | 89.5 |

[a]PPA, positive percent agreement; OPA, overall percent agreement; NP, nasal pharyngeal.

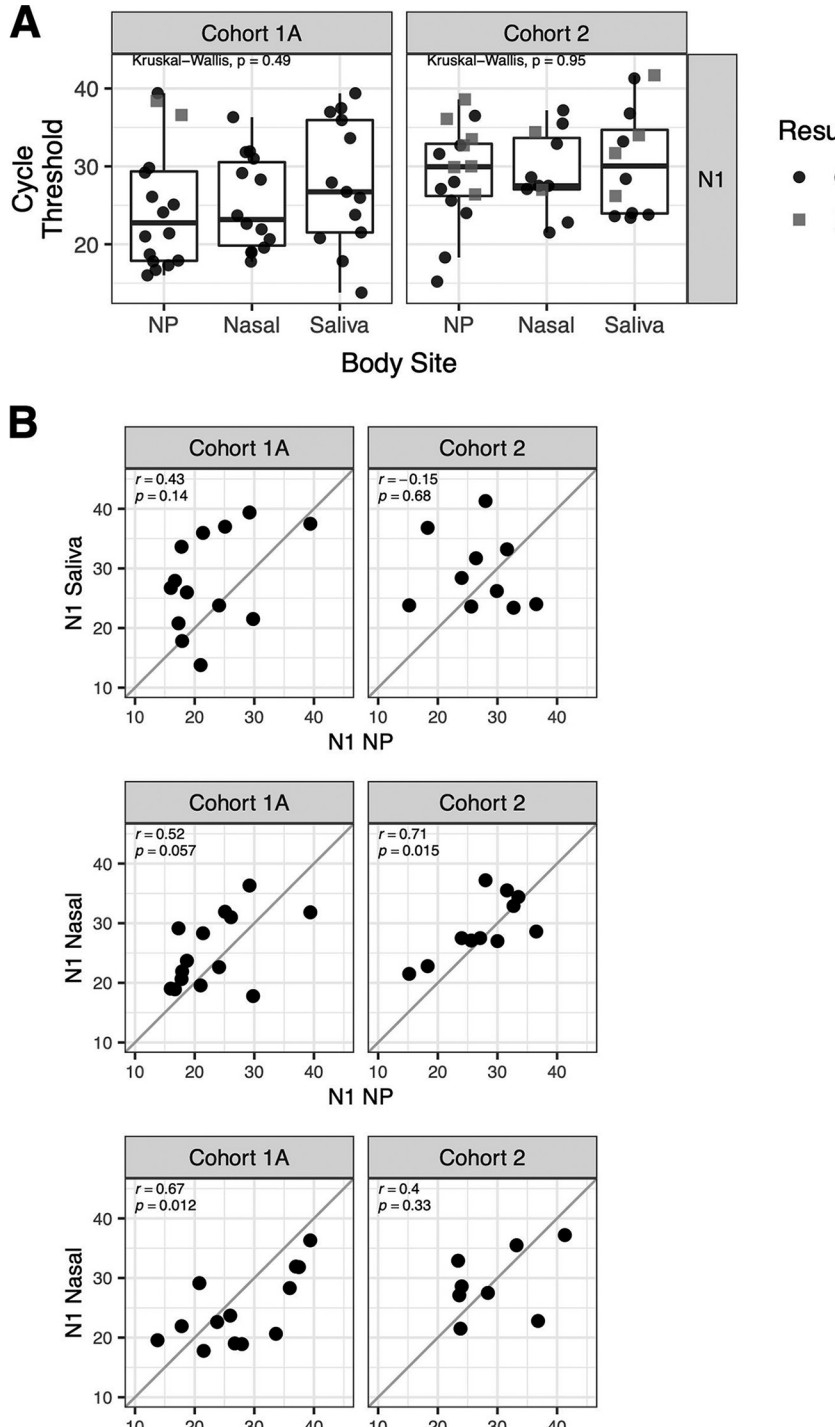

**FIG 2** Cycle thresholds are similar across anatomic sites, and cycle threshold agreement varies by anatomic site and cohort. (A) Box plots showing cycle threshold ($C_T$) values for cohort 1A ($n = 41$) and 2 ($n = 18$) from the CLIA-LDT assay for nasal pharyngeal (NP), nasal, and saliva samples using N1 primers. Black circles are samples with concordance between nasal or saliva samples and nasopharyngeal samples (positive/positive or negative/negative). Gray squares show discordance between nasal or saliva samples and nasopharyngeal samples (positive/negative). Groups are compared using Kruskal-Wallis test by ranks. (B) Scatterplots of $C_T$ values from N1 primers for concordant samples from different sample types: saliva versus nasopharyngeal, nasal versus nasopharyngeal, and nasal versus saliva. Correlation was assessed using Pearson's correlation. The correlation coefficient ($r$) and $P$ value for each comparison are indicated on each plot. See Fig. S1 for analogous analysis of N2 primers and Fig. S2 for analysis of relative viral load.

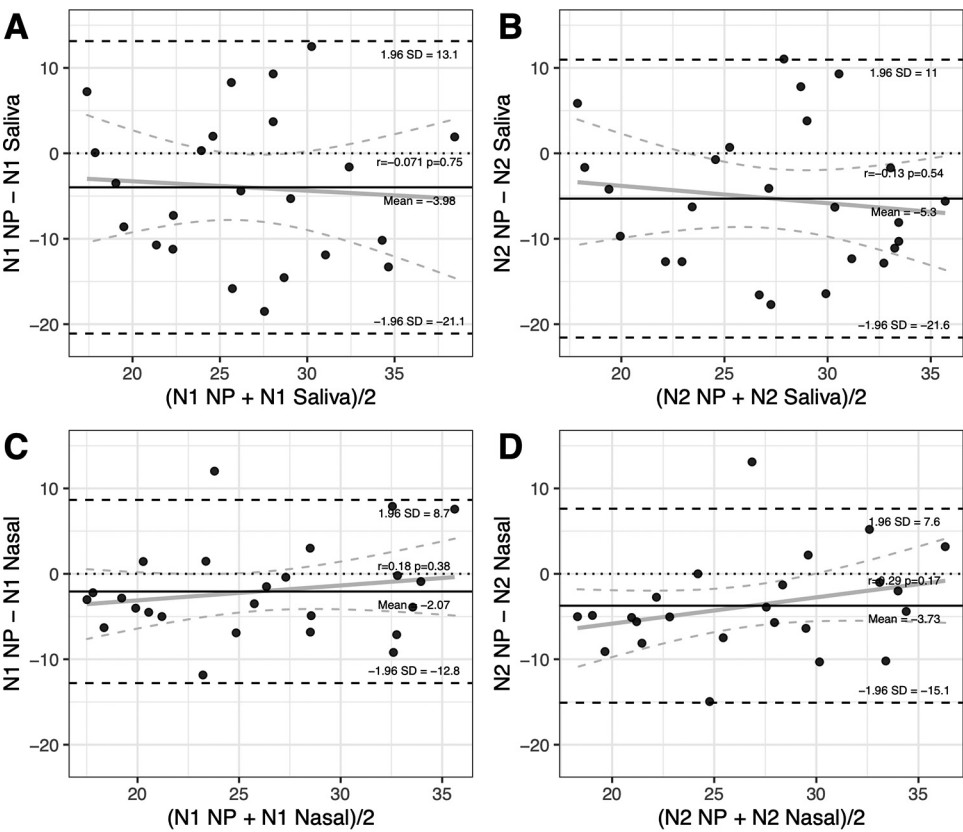

**FIG 3** Bland-Altman analysis reveals minimal proportional bias between saliva and nasal cycle thresholds relative to NP. Bland-Altman (mean-difference) plots show the relationship between average cycle threshold ($C_T$) and the difference between $C_T$ for saliva versus nasopharyngeal (NP) samples for N1 (A) and N2 (B) primers; and nasal versus nasopharyngeal for N1 (C) and N2 (D) primers. The solid black line shows the mean bias for each comparison. Dashed lines represent the limits of agreement (95% confidence interval) around the mean bias (±1.96 standard deviation [SD]). Solid gray line shows the linear relationship between the mean and difference, with dashed gray lines showing the 95% confidence interval for this relationship. Pearson's correlation ($r$) and $P$ value are reported for the correlation between the mean and the difference.

Comparison of N1 (Fig. 2B) and N2 (Fig. S1B) $C_T$ values between sample types from the same patient showed that saliva samples were heterogeneous compared to both nasopharyngeal and nasal samples, generally showing low, nonsignificant correlation. The correlation of nasopharyngeal versus nasal $C_T$ values was relatively tighter, with three of four comparisons trending toward or reaching statistical significance.

Bland-Altmann analysis using samples from cohort 1A and cohort 2 demonstrated a mean bias of −3.98 for N1 target $C_T$ values and −5.3 for N2 target $C_T$ values, indicating average lower $C_T$ values for NP samples than saliva samples (Fig. 3A and B). Comparison of nasal and NP samples demonstrated a smaller mean bias of −2.07 for N1 and −3.73 for N2, again showing lower average $C_T$ values for NP samples (Fig. 3C and D). Heterogeneity around the mean bias was observed with relatively large limits of agreement (95% confidence ranges). There was no significant skewing or proportional difference at early or later $C_T$ values.

**Correlation of patient symptoms with test results.** Symptoms associated with COVID-19 were recorded from the medical record, scored, and used to calculate the probability of COVID [P(Covid)] according to the Menni et al. prediction model (16). Adequate records to calculate P(Covid) were available for 17 of 41 patients in cohort 1A (outpatient), 16 of 19 patients in cohort 1B (ED), and 19 of 20 patients in cohort 2 (inpatient). Two patients in cohort 1A had elevated P(Covid) scores (>0.5) but false-negative saliva and nasal samples (Fig. 4). Interestingly, both were being retested due to persistent symptoms at 2 and 4 weeks, respectively, after onset of their laboratory-confirmed COVID-19. No clear pattern of symptoms within cohort 2 was evident in

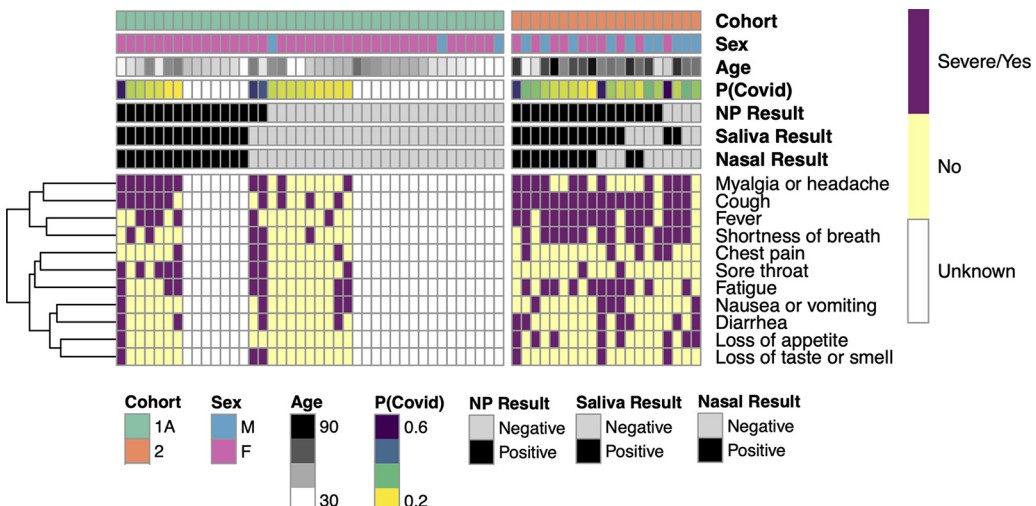

**FIG 4** Symptom heat-map for agreement between positive and negative test results by test method and cohort. Heat map columns are participants and rows are symptoms. Symptom presence is indicated as either severe or yes (purple), no (yellow), or unknown (white). Symptoms are clustered by similarity, and clustering is indicated by the dendrogram (left). Test results are indicated by color as either positive (black) or negative (light gray) for nasopharyngeal (NP), saliva, and nasal samples. Each participant's calculated probability of COVID-19 [P(Covid)] is indicated with higher probabilities shown in purple and lower probabilities in yellow. Participant sex, age, and cohort are also indicated.

relation to the concordant or discordant test results from the three anatomic sites (Fig. 4). In cohort 1B, discrepancies between elevated P(Covid) scores and negative saliva tests (Fig. S3A) occurred in three patients without objective fever or oxygen saturation abnormalities who had complete resolution of subjective symptoms in <48 h. We note that the P(Covid) score is heavily weighted toward loss of taste or smell, a subjective symptom. Overall, saliva and nasal samples demonstrated complete clinical agreement with nasopharyngeal samples in patients with low P(Covid) scores (<0.5) who were tested when initial symptoms developed, suggesting that saliva and nasal samples perform well in the population screening setting.

**Limitations.** This study has a number of limitations due largely to technical and logistical constraints. The study was not intended to be a case-control study but rather was intended to meet the emergency use authorization guidance from the FDA for clinical assessment of test performance with these alternative sample types in comparison to what was considered at the time a gold standard sample for respiratory virus detection. At the time this study was conducted, the guidance was to obtain at least 30 positive and 30 negative samples for clinical assay validation; therefore, there was no power analysis performed prior to study initiation. The study is underpowered to fully address and detect differences in sample type performance, which was further complicated by the unplanned heterogeneity in clinical nasopharyngeal testing in cohort 1. In cohort 1B there is also technical bias between the commercial and CLIA-LDT platforms, as discussed above, that confounds the results and interpretation of that subcohort. The cohorts in this study were not selected with the intention of comprehensively assessing disease prevalence or positive or negative predictive values. Therefore, these results should be interpreted cautiously given small sample sizes and the potential for bias in study cohort selection and uncertain pretest probability or prevalence of COVID-19 in the two cohorts (17). We note that cohort 2 was intentionally enriched for cases in order to assess agreement of the methods in patients with clinically significant COVID-19 diagnosis.

There are several limitations related to the manner in which sampling was carried out. There is a possibility that the order of sampling, with nasopharyngeal performed first, could introduce bias. Nasopharyngeal samples can also detect more historic cases, possibly due to viral integration into the nasopharyngeal epithelium (18). This could

confound results when testing for new-onset disease and could cause an underestimation of the sensitivity of saliva and nasal samples for active infection. Opportunistic collection in emergency departments in cohort 1B, especially during the high-pressure setting of a pandemic, may also have had an impact on the quality of research sample collection and may have impacted the skewed results observed in cohort 1B.

## DISCUSSION

COVID-19 is a challenge to diagnose and treat at the individual patient level and to manage through public health measures at the community level. Alternative sample types, such as patient-acquired saliva and anterior nasal samples, are needed to facilitate high-volume testing when there are shortages of qualified health care workers and necessary supplies to support provider-acquired nasopharyngeal sampling. Our study sought to compare the performance of saliva and anterior nasal samples with standard-of-care NP samples for SARS-CoV-2 detection in different clinical settings. We observed variability in test results dependent on the clinical setting, where samples were collected and the heterogeneous pattern of clinical test routing that was an uncontrolled variable in cohort 1. Our results highlight some of the challenges facing diagnostic characterization of COVID-19 and indicate good performance of saliva and anterior nares (nasal) samples to detect SARS-CoV-2 virus when patients present for outpatient testing early in their disease course.

Variability in laboratory test results is caused by both preanalytical and analytical factors. COVID-19 is a biologically heterogeneous infection: the anatomic distribution of active viral replication and the severity of symptoms compared to viral load are variable between different patients and during each individual's disease course. These are important preanalytical factors impacting test performance on samples from different anatomical sites. A recent systematic review and meta-analysis of saliva samples found that the sensitivity of saliva (83%) was similar to that of nasopharyngeal samples (84%), with diagnostic equivalency highest in the ambulatory setting (19). Additionally, recent preprint data have been released supporting the hypothesis that saliva viral load is a strong clinical predictor of COVID-19 severity (20). These data overall support the important role saliva testing can play in assessing COVID-19.

We observed good PPA in cohort 1A between the testing of banked saliva and nasal samples compared to the clinical nasopharyngeal result. Analytical variation was minimized in this cohort, with all samples tested using the same reverse transcription-PCR (RT-PCR) assay. Saliva and nasal samples showed complete agreement, and only two patients (of 16 total positive) had false-negative saliva and nasal sample results compared to nasopharyngeal results. These patients had previous laboratory-confirmed COVID-19 infections and were being retested due to persistent symptoms. The clinical nasopharyngeal sample for both of these patients had $C_T$ values consistent with viral loads below the 95% confidence limit of detection for the clinical assay. Therefore, the false-negative results in these cases could be due to degradation of the low viral load during the freeze-thaw cycle inherent to the preanalytical study sample handling for this cohort. Considering only patients presenting within the first 10 days of symptom onset ($n = 14$), saliva, nasal, and nasopharyngeal samples had 100% PPA, suggesting that all three anatomic sample sites perform well for initial diagnosis of COVID-19.

Biologic heterogeneity regarding persistence and anatomic distribution of viral replication later in COVID-19 course was apparent in cohort 2 (inpatient setting). An analytical advantage of cohort 2 was the prospective, parallel collection of all three sample types with simultaneous handling and testing on the same platform. In this setting, no single anatomic site provided a positive result for all 18 patients with at least one positive sample type. Nasopharyngeal and saliva samples performed better than nasal samples; however, the combination of an observed, patient-collected nasal and saliva sample together detected the same number of unique positive patients ($n = 16$) as did the provider-collected nasopharyngeal sample. Theoretically, combining patient-collected

saliva and nasal samples into the same collection-stabilization buffer could improve clinical sensitivity.

The data from cohort 1B are difficult to interpret confidently. The unintended use of different molecular tests for the clinical NP test was an analytical confounder discussed above as platform bias. The reported limit of detection for the commercial platforms (250 to 500 viral copies/ml) is similar to the CLIA-LDT (560 copies/ml), but these are not calibrated on the same reference material. Independent clinical quality assurance data from our laboratory comparing replicate testing of nasopharyngeal samples between platforms showed PPA of 90 to 97% (see Tables S4 and S5 in the supplemental material), a level of agreement aligned with published data using standard-of-care nasopharyngeal samples (21–23). This suggests additional preanalytical variables impacted the poor concordance observed in cohort 1B. The clinical vignettes of discrepant results across anatomical sites and testing platforms in cohort 1B highlight the challenges of medical decision-making with limited knowledge of the associations between laboratory test data and the natural biology of SARS-CoV-2 infection.

Our results highlight the importance of clinical judgment and integration of the patient's medical presentation when interpreting individual test results. Understanding how anatomic site, timing of collection in disease course, handling and transport of the specimen, and analytical platform can influence test results is crucially important to making informed medical decisions regarding COVID-19 management (24). Continued study of these factors in diverse clinical settings is necessary for the medical field to improve the response to this global pandemic. Overall, our findings support the conclusion that self-collected saliva testing is effective for COVID-19 detection, especially in early stages of disease progression.

## MATERIALS AND METHODS

**Sample collection, handling, and biobanking.** Nasopharyngeal samples were collected as posterior nasopharynx swabs into viral or universal transport medium by a health care provider for all participants by following standard clinical procedures via sampling through a single nostril. Study saliva and anterior nares (nasal) samples were patient self-collected under direct observation immediately following collection of the nasopharyngeal sample. Participants were instructed to collect 1 ml of saliva before swabbing each nostril. Study participants were given saliva testing kits (item OM-505; DNA Genotek, Ottawa, ON, Canada) and anterior nares (nasal) testing kits (item OCD-100; DNA Genotek, Ottawa, ON, Canada) with written and verbal instructions from monitoring health care workers (Fig. S4). Patients were instructed to spit oral saliva into the collection device; patients were not instructed to expectorate mucous into the sample. Both of these kits deposit the patient-collected sample into a proprietary buffer that inactivates virus and stabilizes nucleic acids.

In cohort 1, liquid saliva samples mixed with buffer, and nasal swabs in buffer were transported at room temperature and stored at −20°C, according to instructions from the manufacturer. Selected research samples were thawed at room temperature immediately prior to analysis on the CLIA-LDT. In cohort 1, nasopharyngeal swab samples were collected, handled, and processed immediately according to the normal course of clinical testing through the health care system (cohort 1A, CLIA-LDT; cohort 1B, commercial assays).

In cohort 2, all samples were collected simultaneously within 48 h of admission. The three sample types were transported together at room temperature to the CLIA-LDT testing facility and processed within 24 h.

**Sample extraction and molecular testing.** An RT-PCR based on primer-probe sets for the SARS-CoV-2 N gene (N1 and N2 targets) and human control RNase P (RP) published by the United States Centers for Disease Control was validated for clinical use (25) by following the regulatory requirements of CLIA and the Federal Drug Administration's Emergency Use Authorization criteria (CLIA-LDT). A sample was reported positive for SARS-CoV-2 if either N1 or N2 viral targets were detected with a $C_T$ of <40 passing data quality control. Negative samples required the internal RP control was detected with a $C_T$ of <38. Extraction of clinical nasopharyngeal samples for testing on this CLIA-LDT assay was performed with either the Qiagen QIAamp viral RNA mini prep (number 52906; Qiagen, Germantown, MD) or the Promega Maxwell RSC viral total nucleic acid kit (number AS1330; Promega, Madison, WI) on the Maxwell RSC instrument per the manufacturer's instructions; bridging studies showed equivalent limits of detection between these two methods (560 viral copies/ml; internal validation data). The sample input volume was 100 μl, and the elution volume was 50 μl for both extraction methods. For saliva and nasal samples obtained in sample buffer, nucleic acid extraction was performed only with the Promega Maxwell method as described above, per the collection device manufacturer's (DNA Genotek) advice. For cohort 1B, nasopharyngeal swab sample testing was performed on the commercial Cepheid Xpert Xpress SARS-CoV-2 test (published limit of detection, 250 copies/ml) or the Diasorin Simplexa COVID-19 direct assay (published limit of detection, 500 copies/ml) by following manufacturer's instructions. See

the supplemental material for additional performance data comparing the CLIA-LDT and commercial assays.

**Medical record review and symptom scoring.** Medical records from telehealth, clinic, or hospital visits were reviewed for relevant symptoms, including loss of taste or smell, shortness of breath, cough, sore throat, fatigue, diarrhea, nausea or vomiting, loss of appetite, chest pain, and myalgia or headache. Physician and nursing notes immediately prior to the initial testing date and over the potentially symptomatic period (approximately 10 to 20 days after testing positive) were reviewed. Subjective symptoms were coded as either present (yes) or absent (no) or as mild, moderate, or severe when the medical record stated severity. In cases where an individual reported at least one symptom; the symptoms that were not reported were considered absent (no). In cases where there was no report of symptoms, symptoms were coded as NA. Objective signs of interest were defined as elevated body temperature (fever) and decreased oxygen saturation based on documentation in the medical record. Data abstraction from the medical records was completed by one study author and reviewed for accuracy by a second author.

**Statistical analysis.** Group mean variation between nasopharyngeal, saliva, and nasal samples were assessed with the Kruskal-Wallis rank sum test. Correlations between methods were assessed using Pearson correlation coefficient. Bland-Altman analysis was used to assess method agreement between saliva or nasal samples relative to nasopharyngeal. PPA was calculated as the proportion of comparative method positives where the test method was positive. Overall percent agreement (OPA) was calculated as the proportion of tests where the test and comparative method agreed. To generate a symptom heat map, symptoms were recoded following the COVID-19 probability score [P(Covid)] prediction equation by Menni et al. (16). Mild cough was recoded as No and only severe fatigue was coded as Yes. Heat map rows are clustered by similarity using the complete linkage method, and columns are sorted by cohort, method concordance, P(Covid), age, and sex. Average cycle threshold was calculated for nasopharyngeal, saliva, and nasal samples as the mean $C_T$ for N1 and N2. Relative viral load was calculated as $[(2^{(RP-N1)}) + (2^{(RP-N2)})]/2$. Data analysis and data visualization was completed using R version 3.4.3 (26) and the following packages: ggplot2 version 3.2.1 (27), cowplot version 0.9.4 (28), tidyverse version 1.3.0 (29), reshape 2 version 1.4.3 (30), pheatmap version 1.0.12 (31), viridis version 0.5.1 (32), RColorBrewer version 1.1-2 (33), and ggpubr 0.2.4 (34).

## SUPPLEMENTAL MATERIAL

Supplemental material is available online only.
**SUPPLEMENTAL FILE 1**, PDF file, 0.4 MB.

## ACKNOWLEDGMENTS

We thank Beth Jorgenson, Sandra Tekmen, Krista Goldsmith, Andrew Snyder, Stephanie McGlone, and Jill Cordes for their efforts in coordination of sample collection. We thank Tyler Bold, Peter Southern, and their laboratory staff for sample management and safety protocols. We acknowledge the significant efforts toward the COVID-19 testing effort made by faculty and staff at the University of Minnesota Genomics Center (Benjamin Auch, Patrick Grady, Darrell Johnson, Ray Watson, Lindsey Gengelbach, Dinesha Walek, Paige Marsolek, Shea Anderson and many others), the Department of Laboratory Medicine & Pathology (Leo Furcht, Patricia Ferrieri, Anthony Killeen, Bharat Thyagarajan, Amy Karger, and Sophie Arbefeville), the Department of Microbiology & Immunology (Ryan Langlois and Ashley Haase), and the Medical School Office of the Dean (Jakub Tolar, Timothy Schacker, Lisa Johnson, and staff). We thank DNA Genotek (Ottawa, Canada) for donating the nasal and saliva kits, and Rafal Iwasiow for general advice and for suggesting useful references. We thank the healthcare staff at the collection sites who took the time to enroll subjects and collect additional samples during the pandemic, and we thank the research subjects for providing additional samples. Finally, we thank the leadership and countless clinical laboratory staff from M Health Fairview, including Kylene Karnuth, Shannon Gascoigne, Michaela Leary, Jessica Gunderson, Klint Kjeldahl, Wendy Walters, and many others.

D.K. and D.M.G. serve as Senior Scientific Advisors to Diversigen, a company involved in the commercialization of microbiome analysis. A.J.J., S.Z., S.L.H., B.H., M.S., R.K., J.D., K.B., S.Y., and A.C.N. have no conflict.

This work was supported by University of Minnesota Office of the Vice President for Research COVID19 rapid response grant number 06 to D.K.

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
