## [Reviewer comments · Microbiology Spectrum]

Microbiology Spectrum

Saliva testing is accurate for early-stage and presymptomatic COVID-19

Abigail Johnson, Shannon Zhou, Suzie Hoops, Benjamin Hillmann, Matthew Schomaker, Robyn Kincaid, Jerry Daniel, Kenneth Beckman, Daryl Gohl, Sophia Yohe, Dan Knights, and Andrew Nelson

Corresponding Author(s): Andrew Nelson, University of Minnesota

Review Timeline:

Submission Date:	April 21, 2021
Editorial Decision:	April 27, 2021
Revision Received:	June 11, 2021
Accepted:	June 14, 2021

Editor: Heba Mostafa

Reviewer(s): The reviewers have opted to remain anonymous.

Transaction Report:

DOI: <https://doi.org/10.1128/Spectrum.00086-21>

April 27, 2021

Dr. Andrew C Nelson
University of Minnesota
Minneapolis

Re: Spectrum00086-21 (Saliva testing is accurate for early-stage and presymptomatic COVID-19)

Dear Dr. Andrew C Nelson:

Thank you for submitting your manuscript to Microbiology Spectrum. As you will see the reviewers support publication of a revised paper. Please revise the paper along the lines suggested by the reviewers. When submitting the revised version of your paper, please provide (1) point-by-point responses to the issues raised by the reviewers as file type "Response to Reviewers," not in your cover letter, and (2) a PDF file that indicates the changes from the original submission (by highlighting or underlining the changes) as file type "Marked Up Manuscript - For Review Only". Please use this link to submit your revised manuscript - we strongly recommend that you submit your paper within the next 60 days or reach out to me. Detailed information on submitting your revised paper are below.

Link Not Available

Sincerely,

Heba Mostafa

Journals Department
Reviewer comments:

Reviewer #1 (Comments for the Author):

The authors present a well conducted study with robust discussion of the findings. The method applied in cohort 1B nicely demonstrates that questionable results can arise, that do not necessarily reflect the reliability of the sample type itself, rather that just because a method is suitable for swabs, does not mean that it will work on other sample types, in this case, saliva.

Major

The authors might like to touch on the fact that NP swabs can detect more historic cases so that this can also confound results when testing for active infection.

More details required on extraction methods - volumes from samples and elution volumes just for clarity so readers know how to replicate good methods, how methods could be altered or what should be avoided.

Minor:

Suggest using nasopharyngeal in place of NP as same word count and remains clear to the reader at all times.

Reviewer #2 (Comments for the Author):

The authors have chosen an important topic, but I find the paper difficult to read and interpret.

Issues requiring clarification:

Cohort 1 consists of both symptomatic and asymptomatic patients. If possible, the authors should provide the number of symptomatic and asymptomatic patients in each of the analyzed subgroups.

It appears that there is significant risk for test platform bias in the testing for Cohort 1B. It would be useful if the authors could provide limit of detection information for the CLIA LDT, and information on the test platforms utilized by the commercial testing facilities, since comparison testing has shown very significant variation in assay performance.

Authors should note the significant risk of bias associated with their case-control design. Ideally, they would have provided a power analysis prior to choosing sample numbers. It appears the study is highly underpowered to detect differences among the three methods.

Authors fail to tell us whether the samples for nasal, NP and saliva testing were obtained at the more or less the same time, although it appears, by comparison with the description of cohort 2, that they were not. This raises the risk for "flow and timing" bias.

Cohort 2 suffers from case selection bias but does not risk test platform bias.

Authors fail to tell us whether the order was randomized. Authors also fail to tell us whether both sides were sampled for NP and nasal sampling and fail to describe whether nasal samples were anterior nares or midturbinate. Authors should also provide, if possible, information on the sample

collection devices for both cohort 1 and 2, and describe the transport buffer.

Details of saliva testing were not provided for either cohort. Authors should make clear whether participants were instructed to cough up mucus into their saliva, instructed NOT to get mucous in their saliva, or not given instruction on this at all.

Analysis

I personally prefer the use of composite reference standards^{1,2} to the approach used by the authors, although their use of PPA is in no sense incorrect. In the absence of 95% confidence intervals around the Bland-Altman regressions, I believe there is a risk that authors are overinterpreting these results. There is no power analysis, and it appears to me that the study is likely underpowered to detect differences in the three sampling methods.

The authors state that "This heterogeneity suggests that seeking a negative COVID-19 molecular test to guide further patient management has limited clinical value, as a negative test from one anatomical site does not rule out the presence of viral RNA in other sites from the same patient." This is true but largely uninformative since repeat testing is rarely used or justifiable for managing patients. Readers could also get the wrong idea about interpretation of negative test results in the ordinary diagnostic setting. For these determinations, the proper approach is to calculate the predictive value of a negative result. In the usual case of a 5-15% prior probability of a positive result, the negative predictive value of these tests will not differ by a significant amount.

1. Baughman AL, Bisgard KM, Cortese MM, Thompson WW, Sanden GN, Strebel PM. Utility of composite reference standards and latent class analysis in evaluating the clinical accuracy of diagnostic tests for pertussis. *Clinical and Vaccine Immunology*, 2008, 15:106-14
2. Tang S, Hemyari P, Canchola JA, Duncan J. Dual composite reference standards (dCRS) in molecular diagnostic research: A new approach to reduce bias in the presence of imperfect reference. *Journal of Biopharmaceutical Statistics*, 2018, 28:951-65

Staff Comments:

Preparing Revision Guidelines

- Point-by-point responses to the issues raised by the reviewers in a file named "Response to Reviewers," NOT IN YOUR COVER LETTER.
- Upload a compare copy of the manuscript (without figures) as a "Marked-Up Manuscript" file.

- Each figure must be uploaded as a separate file, and any multipanel figures must be assembled into one file.
- Manuscript: A .DOC version of the revised manuscript
- Figures: Editable, high-resolution, individual figure files are required at revision, TIFF or EPS files are preferred

For complete guidelines on revision requirements, please see the Instructions to Authors at [link to page]. **Submissions of a paper that does not conform to Microbiology Spectrum guidelines will delay acceptance of your manuscript.**

Due to the SARS-CoV-2 pandemic, our typical 60 day deadline for revisions will not be applied. I hope that you will be able to submit a revised manuscript soon, but want to reassure you that the journal will be flexible in terms of timing, particularly if experimental revisions are needed. When you are ready to resubmit, please know that our staff and Editors are working remotely and handling submissions without delay.

If you would like to submit an image for consideration as the Featured Image for an issue, please contact Spectrum staff.

Response to reviewers:

Reviewer #1 (Comments for the Author):

The authors present a well conducted study with robust discussion of the findings. The method applied in cohort 1B nicely demonstrates that questionable results can arise, that do not necessarily reflect the reliability of the sample type itself, rather that just because a method is suitable for swabs, does not mean that it will work on other sample types, in this case, saliva.

Major

The authors might like to touch on the fact that NP swabs can detect more historic cases so that this can also confound results when testing for active infection.

We agree with this concern, and we have added a description of this to the new "Limitations" section that was added to address this and other reviewer critiques.

More details required on extraction methods - volumes from samples and elution volumes just for clarity so readers know how to replicate good methods, how methods could be altered or what should be avoided.

We have expanded and clarified Materials and Methods, section "Sample extraction and Molecular Testing" regarding extraction methods as requested. Note that clinical testing of nasopharyngeal samples tested on the CLIA-LDT were extracted on either the Qiagen or Promega methods described (due to the reality of clinical supply chain issues from March-June 2020). Input volumes for both procedures were intentionally harmonized to 100 uL and elution volumes to 50 uL to minimize variability in Limit of Detection. Validation data bridging the two extraction methods on the same CLIA-LDT RTqPCR method demonstrated that both generated an equivalent LOD of 560 viral copies per milliliter. The Promega method was recommended by the manufacturer (DNA Genotek) for the proprietary buffer; therefore, this method was used exclusively for the saliva and nasal study samples.

Minor:

Suggest using nasopharyngeal in place of NP as same word count and remains clear to the reader at all times.

We have completed this change throughout the manuscript as recommended.

Reviewer #2 (Comments for the Author):

The authors have chosen an important topic, but I find the paper difficult to read and interpret.

We thank the reviewer for this useful critique and we have made changes to the paper that we believe improve readability. Specifically, we moved discussion of the cohort selection out of the Materials and Methods and into the Results (section “Study Design and Cohort Definition”) to better orient the reader to the study design before presenting the results.

Issues requiring clarification:

Cohort 1 consists of both symptomatic and asymptomatic patients. If possible, the authors should provide the number of symptomatic and asymptomatic patients in each of the analyzed subgroups.

We thank the reviewer for highlighting areas of the study requiring clarification and additional details. For this specific point, we have added text to more clearly describe symptomatic and asymptomatic patients in Cohort 1. Specifically: 1A had 18 of 41 patients with reliable clinical notes, 14 of whom displayed at least one symptom and 4 who were asymptomatic. In 1B, 16 of 19 patients had reliable clinical notes available; 13 were symptomatic and 3 were asymptomatic.

We have updated this information in Results section “Cohort Characteristics” and more specifically indicated clinical nasopharyngeal test results in these 7 asymptomatic patients.

It appears that there is significant risk for test platform bias in the testing for Cohort 1B. It would be useful if the authors could provide limit of detection information for the CLIA LDT, and information on the test platforms utilized by the commercial testing facilities, since comparison testing has shown very significant variation in assay performance.

We agree that the unplanned routing of clinical nasopharyngeal testing during the early phases of the pandemic response led to test platform bias. During the challenges of April-May 2020 when Cohort 1 samples were collected, the routing of clinical testing changed literally day by day, if not hour by hour depending on the number of IVD kits available from Cepheid and Diasorin at our institution (which were typically less than 100 per day). We have included additional quality assurance data comparing the CLIA-LDT with either Cepheid or Diasorin from our institution. In these comparisons, the CLIA-LDT was similarly disadvantaged as in the study: the CLIA-LDT was performed on previously frozen nasopharyngeal samples that had been up-front testing on the commercial instruments. Due to massive supply shortages with these commercial platforms and physical separation of the testing laboratories, it was not possible to do simultaneous sample splits and parallel testing or re-test archived samples from the CLIA-LDT (which was much more scalable than the commercial tests). Nonetheless, those levels of PPA were higher than in the cross-comparison of different anatomical site and test platform. Suggesting again that there are other factors at play.

To address the Reviewer's critique, we have more clearly called attention to that supplemental QA data and we have inserted the published limits of detection in the main body of the manuscript. Specifically:

We have more clearly defined the difference in testing platform for cohort 1A vs 1B in Results section "Study Design and Cohort Definition";

We have acknowledged platform bias and more clearly cited internal QA data provided in the supplemental information including supplemental tables 4 and 5 in Results section "Concordance of Testing on Saliva, Nasal, and NP Nasopharyngeal Samples". Please also note that we review the LOD of the various platforms and describe caveats of that separate QA analysis in the supplemental information;

We again acknowledge platform bias in the newly added "Limitations" section;

And we explicitly state the LOD from our CLIA-LDT validation as well as those published by Diasorin and Cepheid in the Materials and Methods section "Sample Extraction and Molecular Testing."

Authors should note the significant risk of bias associated with their case-control design. Ideally, they would have provided a power analysis prior to choosing sample numbers. It appears the study is highly underpowered to detect differences among the three methods.

We appreciate the reviewer's opinion that the 30+30 study design is underpowered to assess differences among anatomical sample sites, particularly given the confounder of unintended platform bias that occurred in Cohort 1 due to factors beyond the control of the study authors. Samples were collected in April-May 2020 with the original intention of justifying saliva sampling within our institution (with a prospective IRB approval so that we could ultimately share results). Therefore, it was intended to be a retrospective analytical comparison of specimen-type performance meeting FDA EUA guidelines for the scope of a clinical evaluation at that time to justify reasonable inter-operability (these guidelines can be accessed here as of May 26, 2021: <https://www.fda.gov/media/135659/download>).

This has been noted in the Results section "Study design and cohort definition". Cohort 2 was launched after cohort 1 to address a) concerns with staggered analysis of bio-banked samples and b) concerns related to platform bias. Given that the study design was intended to accrue sufficient positive and negative cases to meet FDA EUA guidelines as quickly as possible during the initial phases of the global pandemic, we did not perform a power analysis prior to launching the study. We have added language in the "Limitations" section to indicate that a power analysis was not performed, and that the study is under-powered to detect small differences in the performance of each anatomic sample type.

Authors fail to tell us whether the samples for nasal, NP and saliva testing were obtained at the more or less the same time, although it appears, by comparison with the description of cohort 2, that they were not. This raises the risk for "flow and timing" bias.

We apologize that our methods were not clear on the timing of sample collection from each patient in the different cohort settings. The research saliva and anterior nares samples were collected immediately following the collection of the nasopharyngeal sample for all patients (in both Cohorts 1 and 2), but for Cohort 1 the saliva and anterior nares samples were bio-banked at manufacturer's recommended frozen storage conditions prior to testing versus in Cohort 2 both collection and testing occurred simultaneously. This has been clarified in Materials and Methods section "Sample Collection, Handling, and Biobanking". We have also highlighted in the "Limitations" section that the collection of saliva and nasal swabs only after the collection of NP swabs could have introduced bias.

Cohort 2 suffers from case selection bias but does not risk test platform bias.

We thank the reviewer for this critique and appreciate that cohort 2 (and cohort 1) was enriched for cases. This was by design, as our intention with cohort 2 was to compare the methods on known positive cases, independent of control performance. However, we recognize that enrolling patients who previously tested positive with NP and had clinical symptoms potentially included some patients who were falsely positive or excluded some possible patients who were false negative during screening. This could either bias the results in different directions of sensitivity and specificity. To account for some of this possible bias introduced by sampling only people with previous positive tests, we report the results of retesting all patients with the three methods and using a composite reference approach as suggested by this reviewer. Additionally, we recognize that these patients are not representative of the general population where testing is performed in individuals with a range of symptom severity, and not representative of all hospitalized patients given that they were well enough to participate in self-collection. In the context of this study however, which aimed to quickly evaluate a diagnostic test collection method, this cross-sectional study of clinically presenting, positive Covid-19 status individuals is still beneficial. We have added language to the "Limitations" section to highlight that Cohort 2 was intentionally designed to assess performance of the methods in patients with clinically significant COVID-19 diagnoses.

Authors fail to tell us whether the order was randomized. Authors also fail to tell us whether both sides were sampled for NP and nasal sampling and fail to describe whether nasal samples were anterior nares or midturbinate. Authors should also provide, if possible, information on the sample collection devices for

both cohort 1 and 2, and describe the transport buffer.

The order of anatomic sample collection was not randomized because the saliva and anterior nares samples were considered research/clinical validation and therefore per approval were collected after the clinical nasopharyngeal sample. Participants were instructed to collect saliva before nares samples. The clinical nasopharyngeal swab was taken per routine by sampling the posterior nasopharynx via only one nostril randomly selected by the clinical provider. The nasal samples were anterior nares swabs with both nostrils swabbed by the participant using the same sponge. Images of the collection devices are now included in the supplemental information as Supplemental Figure 4. The transport buffer for the saliva and nasal devices is proprietary. For the clinical nasopharyngeal samples, a variety of viral or universal transport mediums validated for all testing platforms in the institution were used. Please see Materials and Methods section “Sample Collection, Handling, and Biobanking”.

Details of saliva testing were not provided for either cohort. Authors should make clear whether participants were instructed to cough up mucus into their saliva, instructed NOT to get mucous in their saliva, or not given instruction on this at all.

Patients were NOT instructed to expectorate mucous. The OM-505 device from DNA Genotek is a “spit kit” device intended to collect typical oral saliva. We apologize for the lack of clarity and have updated the methods to convey this important information. Please see Materials and Methods section “Sample Collection, Handling, and Biobanking”.

Analysis

I personally prefer the use of composite reference standards [1,2] to the approach used by the authors, although their use of PPA is in no sense incorrect. In the absence of 95% confidence intervals around the Bland-Altman regressions, I believe there is a risk that authors are overinterpreting these results. There is no power analysis, and it appears to me that the study is likely underpowered to detect differences in the three sampling methods.

This is an excellent suggestion, and we have explored the opportunity to apply this approach to our analyses. In Cohort 1, we were constrained to using the clinical nasopharyngeal swab result as the reference because all samples with a negative nasopharyngeal test result also had negative saliva and nasal test results. However, in cohort 2, due to the multiple potential and, at times, conflicting test results between the sample types, there was an opportunity to incorporate the reviewer’s suggestions, and we have reframed the analysis of cohort 2 in comparison to the composite reference standard approach, as reflected in changes to the description of the cohort 2 analysis in

the Results section, "Concordance of Testing on Saliva, Nasal, and Nasopharyngeal Samples." We feel that this has made the paper stronger.

We have added 95% confidence intervals around the Bland-Altman linear regressions and have added a discussion of the power limitations in this study to the new limitations section to highlight that "The study is under-powered to fully address and detect differences in sample type performance".

The authors state that "This heterogeneity suggests that seeking a negative COVID-19 molecular test to guide further patient management has limited clinical value, as a negative test from one anatomical site does not rule out the presence of viral RNA in other sites from the same patient." This is true but largely uninformative since repeat testing is rarely used or justifiable for managing patients. Readers could also get the wrong idea about interpretation of negative test results in the ordinary diagnostic setting. For these determinations, the proper approach is to calculate the predictive value of a negative result. In the usual case of a 5-15% prior probability of a positive result, the negative predictive value of these tests will not differ by a significant amount.

We recognize the potential for this to be confusing to the reader. We have removed this sentence from the publication.

- 1. Baughman AL, Bisgard KM, Cortese MM, Thompson WW, Sanden GN, Strebel PM. Utility of composite reference standards and latent class analysis in evaluating the clinical accuracy of diagnostic tests for pertussis. *Clinical and Vaccine Immunology*, 2008, 15:106-14**
- 2. Tang S, Hemyari P, Canchola JA, Duncan J. Dual composite reference standards (dCRS) in molecular diagnostic research: A new approach to reduce bias in the presence of imperfect reference. *Journal of Biopharmaceutical Statistics*, 2018, 28:951-65**

June 14, 2021

Dr. Andrew C Nelson
University of Minnesota
Minneapolis

Re: Spectrum00086-21R1 (Saliva testing is accurate for early-stage and presymptomatic COVID-19)

Dear Dr. Andrew C Nelson:

Your manuscript has been accepted, and I am forwarding it to the ASM Journals Department for publication. You will be notified when your proofs are ready to be viewed.

Sincerely,

Heba Mostafa
Editor, Microbiology Spectrum
